# Genome-Wide Identification and Expression Analysis of *SWEET* Gene Family in Strawberry

**Riru Tian, Jiayi Xu, Zichun Xu, Jianuo Li and He Li \***

College of Horticulture, Shenyang Agricultural University, Shenyang 110866, China;
tianriru@stu.syau.edu.cn (R.T.); xujiayi@stu.syau.edu.cn (J.X.); xuzichun@stu.syau.edu.cn (Z.X.);
lijianuo@stu.syau.edu.cn (J.L.)
\* Correspondence: lihe@syau.edu.cn

**Abstract:** The Sugars Will Eventually be Exported Transporter (SWEET) is a class of bidirectional sugar transporter that is involved in critical physiological processes such as plant growth and development, and its response to biotic and abiotic stresses. Currently, there are few reports on the *SWEET* gene family in strawberry. In this study, we mined the *SWEET* gene family members in *Fragaria × ananassa* 'Camarosa' and carefully analyzed their molecular features and expression patterns. The results showed that 77 *FanSWEET* genes existed in the *F. × ananassa* 'Camarosa' genome, and the phylogenetic analysis classified them into four sub-groups. Analysis of gene structure, conserved structural domains, and conserved motifs showed that *FanSWEETs* were highly conserved during the evolutionary process. Expression profiling of the 11 *FanSWEET* genes revealed that three members were highly expressed in strawberry fruits, which were presumed to be involved in sugar transport during strawberry fruit ripening. In addition, based on the exogenous sugar-spraying treatment and quantitative real-time PCR analysis, we found that different members responded to different sugar treatments in different response patterns, and their functions in sugar transport need to be further explored. The present study provides a reference for further analysis of the functions of the *SWEET* gene in strawberry.

**Keywords:** strawberry; *SWEET* gene family; bioinformatics analysis; expression analysis

## 1. Introduction

In plants, carbohydrates synthesized through leaf photosynthesis are the main source of energy [1]. As one of the photosynthesis products, saccharides not only can provide carbon skeleton and energy for the growth and development of organisms but also can serve as signaling molecules to participate in various physiological activities and regulate the expression of relevant genes [2,3]. The process of transporting sugar compounds from source leaves to storage organs cannot be independently transported by the concentration difference inside and outside the cell membrane but must be assisted by specific sugar transporters [4,5]. The transport efficiency of sugar transporter proteins determines how much sugar is transported, which, in turn, affects the accumulation of organic matter in the plant [6]. Therefore, sugar transporter proteins are closely related to fruit yield and quality [7].

Currently, sugar transporter proteins in plants are divided into three groups: monosaccharide transporters (MSTs) [8], sucrose transporters (SUTs) [9], and sugars will eventually be exported transporters (SWEETs) [10]. SWEET proteins are a newly discovered family of sugar transporter proteins [11], which are widely found in prokaryotes, animals, and plants [12,13]. They are capable of energy-independent, bidirectional transport of sugars utilizing the concentration difference between sugars inside and outside the cell [14]. Most SWEET proteins in eukaryotes contain seven transmembrane domains (TMs), which are formed by the repeating tandem of two 3-TM units located at the N-terminal and C-terminal ends, respectively, which are joined by one TM helix to form a 3-1-3 structure [10]. These

SWEET proteins in plants belong to the MtN3/saliva family (PF03083) or the PQ-Loop family (PF04193) within the MtN3-like family and are mostly localized to the plasma membrane [15].

There are numerous members of the *SWEET* gene family in higher plants, and studies on the *SWEET* gene family have been reported: There are 17, 21, 24, 29, 34, 27, 17, 27, 16, 22, 20, 19, 23, 25, and 25 *SWEETs* identified in *Arabidopsis thaliana* [10], rice [16], maize [17], tomato [18], *Brassica rapa* [19], garlic [20], grapes [21], apples [22], litchi [23], watermelon [24], longan [25], jujube [7], *Bletilla striata* [26], *Dendrobium officinale* [27], and rose [28], respectively. The members of the *SWEET* gene family have the specificity of space–time and tissue expression, and there are obvious differences in the types of transported sugar [29]. The phylogenetic analysis of SWEET proteins in plants divides them into four subclasses, and different subclasses of SWEET proteins are closely related to the relative selection of monosaccharides and disaccharides [4]. SWEET proteins of clade III mainly transport sucrose [30], while the other three clades mainly transport monosaccharides including glucose, fructose, and galactose [31]. Experimental studies have found that AtSWEET4/5/8/16 in *A. thaliana* can decompose glucose [29,32,33], AtSWEET16/17 are involved in fructose efflux [32,34], and AtSWEET9/11/12/13/14/15/16 can translocate sucrose [13,33,35,36]. OsSWEET11 is involved in the transport of sucrose in early rice glumes [37]. Rice OsSWEET3a is involved in glucose translocation to leaves during early glume development [38]. And OsSWEET5 is a galactose transporter protein [39]. In tomato, *SlSWEET1a* is highly expressed in young leaf veins and regulates glucose accumulation in thin-walled cells [40]. In addition, it has been shown that SWEET proteins in plants are involved in not only sugar transport [34] but also ion transport [41], maturation senescence [42], plant–pathogen interactions [21], biotic and abiotic stresses, and other important processes [43].

The octoploid strawberry (*Fragaria* × *ananassa* Duch.) is currently the main cultivar and is popular for its reddish appearance, delicious flavor, and rich nutritional value [44]. Sugar accumulation is the key to the formation of fruit quality, but the sugar content of some strawberry cultivars is relatively low, which limits the development of strawberry industry to some extent. SWEET is a key transport protein for transporting sugar, and analyzing the relationship between sugar and SWEET transport is helpful to improve fruit quality. To date, few studies have been conducted on the *SWEET* gene family in strawberry. Therefore, we performed a genome-wide analysis of *SWEET* and characterized the expression patterns of *SWEET*, which will help to further explore the roles of *SWEET* genes in the growth and development of strawberry.

## 2. Materials and Methods

### 2.1. Identification and Physicochemical Properties Analysis of the SWEET Gene Family

The *Fragaria* × *ananassa* Camarosa Genome v1.0.a2 (Re-annotation of v1.0.a1) file as well as the genome annotation file reannotated were downloaded from the GDR database (https://www.rosaceae.org/ (accessed on 3 September 2022)) [45]. The SWEET Hidden Markov Model (HMM) PF03083 was used as a template to download the conserved structural domain data of this family in the Pfam database (http://pfam.xfam.org/ (accessed on 22 June 2022)). With MtN3-slv as a seed model, strawberry proteins were searched in TBtools software (version 2.019) (https://github.com/CJ-Chen/TBtools/releases (accessed on 22 June 2022)). The screening threshold was set to E-value $< 1 \times 10^{-10}$, and the candidate genes encoding SWEET transporters were preliminarily obtained. The protein sequences of the candidate gene family members were then obtained from the strawberry genome-wide database in TBtools software (version 2.019). Each SWEET protein sequence was secondarily screened on the Pfam website to remove genes that did not contain the known MtN3/saliva and PQ-loop domain. The SWEET family members of octoploid 'Camarosa' strawberry were named according to their homology to *A. thaliana* as compared on the Phytozome website (https://phytozome-next.jgi.doe.gov/ (accessed on 22 June 2022)).

The SWEET family members identified by the screening were submitted to ExPASy (https://web.Expasy.Org/compute_pi/ (accessed on 30 July 2022)) for predicting protein molecular weight (MW) and theoretical isoelectric point (pI). The ProtParam (https://web.expasy.org/protparam/ (accessed on 30 July 2022)) website was utilized to determine the average hydrophobicity index of proteins. The online tool TMHMM-2.0 (https://services.Healthtech.Dtu.Dk/service.Php?TMHMM-2.0 (accessed on 30 July 2022)) was used for protein transmembrane helix analysis. The protein instability index and aliphatic index were analyzed in TBtools software (version 2.019). The subcellular localization of each family member was predicted using WoLF PSORT (https://wolfpsort.Hgc.Jp/ (accessed on 26 November 2022)).

### 2.2. Phylogenetic Tree Construction

The seventeen known genetic sequences of *AtSWEETs* were downloaded from the National Center for Biotechnology Information (NCBI) database (http://ncbi.nlm.nih.gov/ (accessed on 7 October 2022)). Homologous relationships of strawberry SWEET proteins were compared on the Phytozome online website (https://phytozome-next.jgi.doe.gov/blast-search (accessed on 7 October 2022)) using *A. thaliana* as the comparison target. The SWEET protein sequences of *F. × ananassa* and *Arabidopsis* were aligned with ClustalW, and the alignment was imported into MEGA11 (https://www.megasoftware.net/ (accessed on 14 October 2022)) to create a phylogenetic tree. And it was constructed using the neighbor-joining (NJ) method, and the bootstrap value was set as 1000. Evolview (https://evolgenius.info//evolview-v2/#mytrees/SHOWCASES/showcase%2001 (accessed on 20 September 2022)) online tool was used to beautify the evolutionary tree.

### 2.3. Chromosome Location and Gene Structure Analysis

The position of the *SWEET* gene family on the chromosome has been mapped using MG2C online tool (http://mg2c.iask.in/mg2c_v2.1/ (accessed on 15 August 2022)). It has been reported that when the amino acid identity of the two sequences is more than 80%, the gene alignment coverage is more than 0.75, and the E expectation value is less than $1 \times 10^{-10}$, the two genes are considered as a pair of replicated genes [16]. When two genes are located on the same chromosome and the interval is less than 100 kb, they are tandem genes [46]. The exon–intron structure was mapped according to the 'Camarosa' strawberry genome annotation file using TBtools software (version 2.019).

### 2.4. Analysis of Conserved Domains and Conserved Motifs

Prediction of conserved structural domains of SWEET proteins was performed using the NCBI website and visualized by TBtools software (version 2.019). Conserved motifs were analyzed using the online software MEME (https://meme-suite.org/meme/tools/meme (accessed on 7 October 2022)) for 'Camarosa' strawberry SWEETs proteins. The parameters were set as follows: the number of replicates was zero or one, the number of motifs was limited to 7, the minimum length of motifs was 10, and the maximum length was 50.

### 2.5. Analysis of Promoter Cis-Acting Elements

According to the genome annotation file, the first 2000bp sequence of the start codon of the *SWEET* gene was extracted in TBtools software (version 2.019). The online software Plant CARE (http://bioinformatics.psb.ugent.be/webtools/plantcare/html/ (accessed on 9 August 2023)) was used to predict the cis-acting elements of the *SWEET* gene, and the data were filtered and organized to visualize the promoter positions.

### 2.6. Plant Materials and Sugar Treatment

The octoploid strawberry (*Fragaria × ananassa*) cultivar 'Yanli' was used as the plant material in this experiment, which was grown in a greenhouse under natural conditions in ShenYang (Liaoning, China; 41° N, 123° E). A total of 20 plants were used for tissue

expression detection, and 90 plants were used for exogenous sugar treatment experiment. The whole strawberry plant was sprayed with 0.1 mol·L$^{-1}$ and 0.2 mol·L$^{-1}$ solutions of glucose, fructose, sucrose, and mixed sugar (composed of glucose, fructose and sucrose with the same concentration) respectively. The treatment code is shown in Table 1. Subsequently, different sugar solutions were sprayed every 3 d for five treatments. Fruit samples were collected 3 days after the last treatment. The samples used for tissue-specific expression analysis were small green fruit stage (SG), big green fruit stage (BG), white fruit stage (W), turning stage (T), red fruit stage (R), and mature leaves (L). All samples were quickly frozen in liquid nitrogen after collection and stored at −80 °C.

**Table 1.** Types and concentrations for the exogenous sugar treatment.

| Treatment Code | Treatment Combination |
| --- | --- |
| CK | Water |
| 0.1 G | 0.1 mol·L$^{-1}$ glucose |
| 0.1 F | 0.1 mol·L$^{-1}$ fructose |
| 0.1 S | 0.1 mol·L$^{-1}$ sucrose |
| 0.1 MS | 0.1 mol·L$^{-1}$ mixed sugar |
| 0.2 G | 0.2 mol·L$^{-1}$ glucose |
| 0.2 F | 0.2 mol·L$^{-1}$ fructose |
| 0.2 S | 0.2 mol·L$^{-1}$ sucrose |
| 0.2 MS | 0.2 mol·L$^{-1}$ mixed sugar |

*2.7. RT-qPCR Analysis of SWEETs*

The total RNA of the above samples was extracted using an RNA extraction kit (TIANGEN, Beijing, China) according to the instructions. And the synthesis of the cDNA was performed using a reverse transcription kit (TaKaRa, Dalian, China). RT-qPCR was carried out using a SYBR Green PCR Master Mix Kit (TaKaRa, Dalian, China) on the CFX96 Real-Time PCR System (Applied Bio-systems, Foster City, CA, USA). The reaction system was as follows: 0.5 µL of template cDNA, 0.5 µL of each upstream and downstream primer, 5 µL of UltraSYBR mixture, and 3.5 µL of ddH$_2$O were added to the reaction system. The qPCRs consisted of a hold at 95 °C for 10 min, then 40 cycles of 95 °C for 15 s and 60 °C for 1 min, and finally 95 °C for 15 s. The Fve26s gene was used as the internal control. The normalized date was processed with TBtools software (version 2.019) and plotted as a heatmap to visualize the changes in *SWEET* gene expression. Primer sequences used for qPCR are shown in Table S1. Three individual samples were used for each treatment, and three biological replicate analyses were also performed.

**3. Results**

*3.1. Identification and Physicochemical Properties Analysis of SWEET of F. × ananassa*

A total of 77 family members of *SWEETs* were excavated in the *F. × ananassa* using haplotype analysis (Table 2). The nomenclature of these strawberry SWEET following a phylogenetic analysis with SWEET homologs from *Arabidopsis* was conducted, which was named *FanSWEET1a~FanSWEET17h*. The minimum number of transmembrane domains in *FanSWEET* was only 2 (*FanSWEET15d*), the maximum contained 14 (*FanSWEET10c*), and 47 members contained 7 TMs, accounting for 61.04% of the total. The theoretical isoelectric points ranged from 5.00 to 9.83, and eight members were acidic proteins, most of which were found in clade III, while all members of clade II were basic proteins. All *FanSWEETs* were hydrophobic proteins. A total of 52 members were stable proteins (instability index < 40), and 25 were unstable proteins. About 80.52% of the members were subcellularly localized at the plasma membrane.

**Table 2.** Physicochemical characteristics of the *SWEET* gene family in *F. × ananassa*.

| Gene ID | Gene Name | Number of Amino Acid (aa) | Number of Predicted TMs | Molecular Weight (kD) | Theoretical pI | Average Hydrophobicity Index | Instability Index | Aliphatic Index | Subcellular Localization |
|---|---|---|---|---|---|---|---|---|---|
| FxaC_6g17780.t1 | *FanSWEET1a* | 249 | 7 | 27.327 | 9.37 | 0.612 | 36.92 | 106.02 | plas |
| FxaC_5g24000.t1 | *FanSWEET1b* | 243 | 7 | 26.752 | 9.28 | 0.696 | 37.28 | 107.86 | plas |
| FxaC_8g24010.t1 | *FanSWEET1c* | 249 | 7 | 27.236 | 9.26 | 0.616 | 36.54 | 104.86 | plas |
| FxaC_7g12850.t1 | *FanSWEET1d* | 283 | 7 | 30.920 | 9.25 | 0.578 | 40.37 | 104.66 | plas |
| FxaC_7g21380.t1 | *FanSWEET1e* | 184 | 5 | 19.885 | 5.23 | 0.770 | 26.62 | 104.84 | plas |
| FxaC_5g14330.t1 | *FanSWEET1f* | 206 | 3 | 22.683 | 9.61 | 0.524 | 34.15 | 95.00 | chlo |
| FxaC_9g37270.t1 | *FanSWEET2a* | 235 | 7 | 26.246 | 9.04 | 0.865 | 45.42 | 121.15 | plas |
| FxaC_10g18570.t1 | *FanSWEET2b* | 235 | 7 | 26.305 | 9.02 | 0.837 | 45.58 | 118.64 | plas |
| FxaC_12g32490.t1 | *FanSWEET2c* | 235 | 7 | 26.179 | 8.93 | 0.838 | 41.80 | 121.15 | plas |
| FxaC_9g48290.t1 | *FanSWEET2d* | 234 | 7 | 26.344 | 8.95 | 0.935 | 38.05 | 130.77 | vacu |
| FxaC_12g43600.t1 | *FanSWEET2e* | 234 | 7 | 26.314 | 9.03 | 0.965 | 38.05 | 131.97 | vacu |
| FxaC_10g00470.t1 | *FanSWEET2f* | 234 | 7 | 26.300 | 9.03 | 0.956 | 38.88 | 131.58 | vacu |
| FxaC_11g04070.t1 | *FanSWEET2g* | 167 | 5 | 18.773 | 8.95 | 1.034 | 33.96 | 139.40 | vacu |
| FxaC_28g03100.t1 | *FanSWEET3a* | 246 | 6 | 27.697 | 8.82 | 0.539 | 42.52 | 112.07 | plas |
| FxaC_26g05400.t1 | *FanSWEET3b* | 257 | 7 | 28.775 | 9.13 | 0.581 | 40.85 | 115.21 | plas |
| FxaC_27g47490.t1 | *FanSWEET3c* | 257 | 7 | 28.869 | 8.99 | 0.538 | 39.72 | 111.44 | plas |
| FxaC_27g50800.t1 | *FanSWEET3d* | 257 | 7 | 28.869 | 8.99 | 0.538 | 39.72 | 111.44 | plas |
| FxaC_25g53100.t1 | *FanSWEET3e* | 257 | 7 | 28.914 | 9.14 | 0.549 | 41.56 | 112.57 | plas |
| FxaC_25g61070.t1 | *FanSWEET3f* | 152 | 4 | 17.099 | 8.71 | 0.474 | 43.96 | 108.22 | chlo |
| FxaC_22g04470.t1 | *FanSWEET4a* | 241 | 7 | 26.733 | 8.80 | 0.761 | 29.47 | 116.76 | plas |
| FxaC_21g05610.t1 | *FanSWEET4b* | 477 | 13 | 53.287 | 9.11 | 0.723 | 32.13 | 114.05 | plas |
| FxaC_22g00450.t1 | *FanSWEET4c* | 242 | 7 | 26.846 | 8.80 | 0.778 | 27.97 | 117.48 | plas |
| FxaC_22g00340.t1 | *FanSWEET4d* | 243 | 7 | 27.328 | 9.24 | 0.740 | 30.76 | 123.42 | plas |
| FxaC_23g47770.t1 | *FanSWEET4e* | 243 | 7 | 27.371 | 9.16 | 0.738 | 28.82 | 121.81 | plas |
| FxaC_21g05660.t1 | *FanSWEET4f* | 255 | 7 | 28.694 | 9.06 | 0.783 | 27.81 | 127.14 | plas |
| FxaC_21g05780.t1 | *FanSWEET4g* | 235 | 6 | 26.664 | 9.39 | 0.620 | 34.21 | 121.83 | plas |
| FxaC_14g26270.t1 | *FanSWEET4h* | 169 | 4 | 18.959 | 8.91 | 0.790 | 30.09 | 122.72 | plas |
| FxaC_28g27160.t1 | *FanSWEET4i* | 167 | 5 | 18.780 | 8.21 | 0.763 | 27.20 | 121.92 | plas |
| FxaC_21g24240.t1 | *FanSWEET5a* | 235 | 7 | 26.437 | 9.08 | 0.643 | 31.28 | 121.87 | plas |
| FxaC_23g02850.t1 | *FanSWEET5b* | 248 | 6 | 28.129 | 9.03 | 0.603 | 31.79 | 118.23 | plas |
| FxaC_22g20990.t1 | *FanSWEET5c* | 236 | 7 | 26.530 | 8.93 | 0.624 | 32.05 | 120.13 | plas |
| FxaC_18g40990.t1 | *FanSWEET5d* | 235 | 7 | 26.404 | 9.56 | 0.757 | 45.10 | 125.62 | plas |
| FxaC_19g05870.t1 | *FanSWEET5e* | 117 | 3 | 13.318 | 9.10 | 0.542 | 54.48 | 125.73 | plas |

**Table 2.** *Cont.*

| Gene ID | Gene Name | Number of Amino Acid (aa) | Number of Predicted TMs | Molecular Weight (kD) | Theoretical pI | Average Hydrophobicity Index | Instability Index | Aliphatic Index | Subcellular Localization |
|---|---|---|---|---|---|---|---|---|---|
| FxaC_17g05070.t1 | *FanSWEET5f* | 237 | 7 | 26.618 | 9.37 | 0.732 | 44.45 | 124.98 | plas |
| FxaC_17g07490.t1 | *FanSWEET5g* | 237 | 7 | 26.618 | 9.37 | 0.732 | 44.45 | 124.98 | plas |
| FxaC_20g06160.t1 | *FanSWEET5h* | 237 | 7 | 26.648 | 9.28 | 0.723 | 47.35 | 124.56 | plas |
| FxaC_24g43890.t1 | *FanSWEET5i* | 324 | 8 | 37.062 | 9.67 | 0.289 | 40.99 | 103.12 | plas |
| FxaC_17g12530.t1 | *FanSWEET7a* | 253 | 7 | 28.086 | 9.54 | 0.630 | 32.42 | 120.95 | plas |
| FxaC_17g15900.t1 | *FanSWEET7b* | 253 | 7 | 27.987 | 9.45 | 0.646 | 32.08 | 120.95 | plas |
| FxaC_19g11570.t1 | *FanSWEET7c* | 253 | 6 | 28.122 | 9.45 | 0.634 | 33.46 | 122.49 | plas |
| FxaC_20g12290.t1 | *FanSWEET7d* | 326 | 8 | 36.329 | 9.11 | 0.525 | 42.70 | 118.68 | E.R. |
| FxaC_18g34660.t1 | *FanSWEET7e* | 253 | 7 | 28.064 | 9.57 | 0.641 | 33.35 | 123.28 | plas |
| FxaC_23g42760.t1 | *FanSWEET9a* | 257 | 7 | 28.984 | 9.57 | 0.609 | 34.09 | 111.56 | plas |
| FxaC_22g65080.t1 | *FanSWEET9b* | 568 | 8 | 62.638 | 8.78 | 0.235 | 37.72 | 100.11 | plas |
| FxaC_21g70630.t1 | *FanSWEET9c* | 823 | 8 | 91.383 | 8.50 | 0.085 | 34.67 | 97.96 | plas |
| FxaC_24g06130.t1 | *FanSWEET9d* | 850 | 9 | 94.782 | 8.62 | 0.086 | 34.22 | 98.28 | plas |
| FxaC_22g71200.t1 | *FanSWEET9e* | 537 | 8 | 59.333 | 8.36 | 0.315 | 40.39 | 102.25 | plas |
| FxaC_5g10910.t1 | *FanSWEET9f* | 288 | 7 | 32.130 | 8.70 | 0.651 | 29.99 | 119.06 | chlo |
| FxaC_7g24770.t1 | *FanSWEET9g* | 288 | 7 | 32.100 | 8.38 | 0.638 | 26.18 | 118.09 | plas |
| FxaC_8g17321.t1 | *FanSWEET9h* | 288 | 7 | 31.981 | 8.69 | 0.687 | 25.12 | 120.45 | plas |
| FxaC_5g10940.t1 | *FanSWEET9i* | 191 | 6 | 21.197 | 9.83 | 0.931 | 36.56 | 123.98 | vacu |
| FxaC_8g17360.t1 | *FanSWEET9j* | 550 | 13 | 61.468 | 8.43 | 0.651 | 33.28 | 116.96 | plas |
| FxaC_7g28600.t1 | *FanSWEET9k* | 284 | 6 | 31.639 | 9.44 | 0.558 | 38.37 | 113.31 | chlo |
| FxaC_5g10930.t1 | *FanSWEET9l* | 261 | 5 | 28.760 | 6.25 | 0.739 | 31.65 | 121.42 | vacu |
| FxaC_7g24760.t1 | *FanSWEET9m* | 266 | 5 | 29.186 | 8.86 | 0.654 | 33.54 | 116.58 | vacu |
| FxaC_7g26140.t1 | *FanSWEET9n* | 267 | 6 | 29.272 | 9.02 | 0.649 | 34.73 | 116.14 | vacu |
| FxaC_7g24771.t1 | *FanSWEET9o* | 271 | 7 | 30.236 | 9.33 | 0.696 | 40.65 | 118.75 | plas |
| FxaC_8g17320.t1 | *FanSWEET9p* | 297 | 7 | 33.043 | 8.82 | 0.576 | 39.26 | 113.91 | plas |
| FxaC_5g10900.t1 | *FanSWEET9q* | 297 | 7 | 33.065 | 8.96 | 0.596 | 35.65 | 113.94 | plas |
| FxaC_6g29720.t1 | *FanSWEET9r* | 100 | 2 | 11.202 | 5.00 | 0.409 | 36.93 | 97.50 | golg_plas |
| FxaC_18g37560.t1 | *FanSWEET10a* | 292 | 7 | 33.036 | 8.45 | 0.744 | 44.27 | 124.14 | plas |
| FxaC_19g09170.t1 | *FanSWEET10b* | 292 | 7 | 33.020 | 8.45 | 0.738 | 44.59 | 124.14 | plas |
| FxaC_20g09320.t1 | *FanSWEET10c* | 652 | 14 | 73.570 | 8.70 | 0.504 | 45.95 | 114.34 | plas |
| FxaC_19g09171.t1 | *FanSWEET14* | 310 | 7 | 34.655 | 7.67 | 0.445 | 46.05 | 113.45 | plas |
| FxaC_14g15350.t1 | *FanSWEET15a* | 305 | 7 | 33.936 | 5.58 | 0.527 | 32.14 | 113.44 | plas |
| FxaC_15g15880.t1 | *FanSWEET15b* | 305 | 7 | 33.941 | 5.76 | 0.509 | 34.24 | 111.21 | plas |
| FxaC_13g22660.t1 | *FanSWEET15c* | 305 | 7 | 34.008 | 5.69 | 0.493 | 32.64 | 111.21 | plas |

**Table 2.** *Cont.*

| Gene ID | Gene Name | Number of Amino Acid (aa) | Number of Predicted TMs | Molecular Weight (kD) | Theoretical pI | Average Hydrophobicity Index | Instability Index | Aliphatic Index | Subcellular Localization |
|---------|-----------|---------------------------|-------------------------|-----------------------|----------------|------------------------------|-------------------|-----------------|--------------------------|
| FxaC_14g16350.t1 | *FanSWEET15d* | 95 | 2 | 10.864 | 9.05 | 0.459 | 49.18 | 98.42 | E.R. |
| FxaC_13g23610.t1 | *FanSWEET15e* | 274 | 6 | 30.778 | 5.36 | 0.306 | 33.95 | 105.29 | plas |
| FxaC_13g27850.t1 | *FanSWEET17a* | 241 | 7 | 26.633 | 7.67 | 0.686 | 42.50 | 111.29 | plas |
| FxaC_13g37330.t1 | *FanSWEET17b* | 241 | 7 | 26.647 | 7.67 | 0.686 | 43.86 | 111.29 | plas |
| FxaC_15g18770.t1 | *FanSWEET17c* | 260 | 5 | 28.991 | 9.19 | 0.332 | 45.12 | 103.92 | plas |
| FxaC_26g29920.t1 | *FanSWEET17d* | 235 | 7 | 25.883 | 8.59 | 0.830 | 38.41 | 125.70 | plas |
| FxaC_28g28980.t1 | *FanSWEET17e* | 235 | 7 | 25.883 | 8.59 | 0.829 | 37.14 | 125.28 | plas |
| FxaC_27g11460.t1 | *FanSWEET17f* | 235 | 7 | 25.927 | 7.76 | 0.806 | 37.43 | 124.85 | vacu |
| FxaC_27g17530.t1 | *FanSWEET17g* | 235 | 7 | 25.869 | 8.59 | 0.829 | 37.14 | 125.28 | plas |
| FxaC_25g13450.t1 | *FanSWEET17h* | 190 | 5 | 20.806 | 6.55 | 0.861 | 36.31 | 128.32 | vacu |

Plas, plasma membrane; chlo, chloroplast; vacu, vacuole; E.R., endoplasmic reticulum; golg_plas, Golgi apparatus.

### 3.2. Phylogenetic Tree Construction of F. × ananassa SWEET Family

To better understand the evolutionary origin and function of the strawberry *SWEET* genes, a phylogenetic tree of the SWEETs was constructed based on the amino acid sequences of AtSWEETs (Figure 1). Phylogenetic results revealed that the evolutionary relationship between 'Camarosa' strawberry and the *A. thaliana* SWEET family was consistent. And it could be classified into four clades, in which clade I possessed 19 *FanSWEETs*, clade II possessed 23 *FanSWEETs*, clade III possessed 27 *FanSWEETs*, and clade IV possessed 8 *FanSWEETs*. Although more SWEET members were identified in *F. × ananassa* than in *Arabidopsis*, the homologous proteins of AtSWEE6/8/11/12/13/16 have not been identified in strawberry.

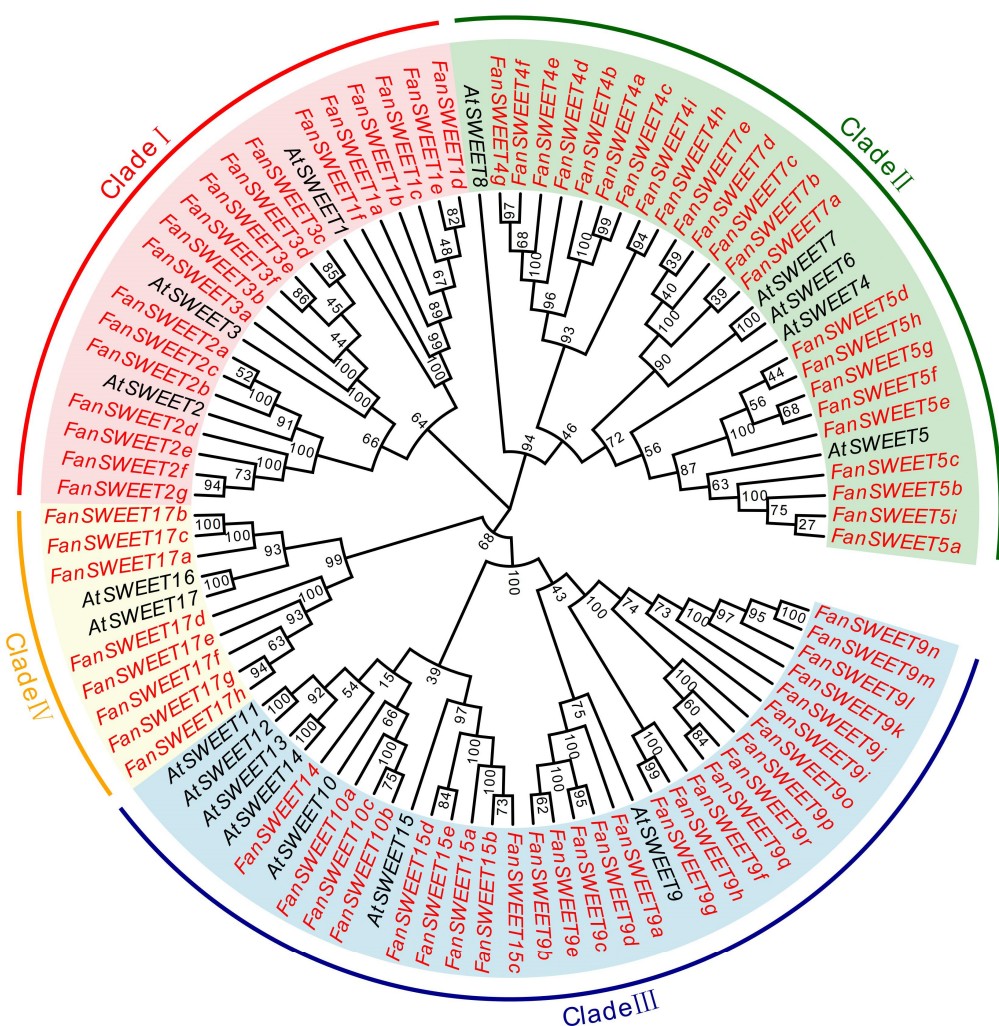

**Figure 1.** Phylogenetic tree of SWEET family of *F. × ananassa* and *A. thaliana*. All SWEET proteins are divided into four clades (I, II, III, and IV), which are represented by different colored backgrounds. At stands for *A. thaliana*, Fan stands for *F. × ananassa*, and *FanSWEET* proteins are marked in red font.

### 3.3. Chromosome 'Location of SWEET Family in F. × ananassa

According to the chromosome location (Figure 2), there is no *FanSWEET* gene in the first set of homologous chromosomes of *F. × ananassa*, and 19, 7, 9, 14, 16, and 12 *FanSWEET* genes are unevenly distributed in the other six sets of homologous chromosomes. The genes on the second and fourth chromosomes are mostly distributed in the middle, while the genes on the other four chromosomes are mostly distributed at both ends. A total of 76 pairs of fragment replication genes and 6 groups of tandem gene clusters were found in 77 *FanSWEETs*. Among them, *FanSWEET4f-FanSWEET4g* located on Chr 6-1 chromosome is a pair of tandem repeat genes.

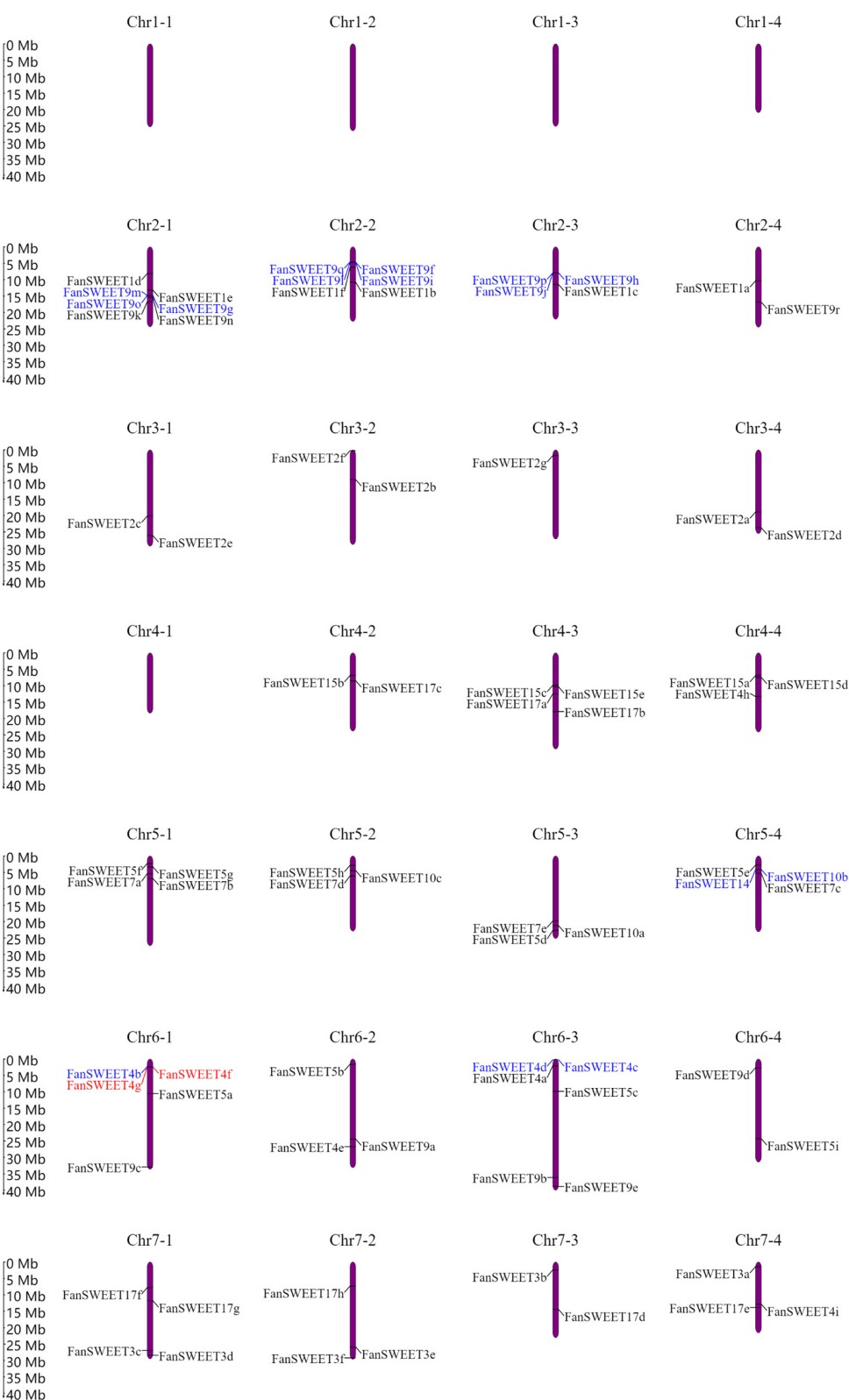

**Figure 2.** Chromosome localization of *SWEET* family members in *F. × ananassa*. The chromosome number is shown above each chromosome. Purple vertical bars with different lengths indicate *F. × ananassa* chromosomes, black short lines indicate the position of each *FanSWEET*, blue fonts indicate tandem genes, and red fonts indicate tandem repetitive genes. The scale bars beside the chromosome indicate the length of megabases (Mb). The arrows show the transcription directions of *FanSWEET* genes.

### 3.4. Analysis of Conserved Domains and Motifs of the SWEET Family in F. × ananassa

By conservative domain analysis (Figure 3), it is found that members of the SWEET family generally have two typical domains, the MtN3_slv or PQ-loop superfamily. Meanwhile, *FanSWEET1e/2g/3f/4h/4i/5e/9l/9m/9n/9r/15d/17c* and 17h have only one structural domain, and *FanSWEET4b*, *FanSWEET9j*, and *FanSWEET10c* have four SWEET family typical structural domains. *FanSWEET* members homologous to AtSWEET2 and AtSWEET3 only have the PQ-loop superfamily structural domains and not the MtN3_slv structural domain. Furthermore, *FanSWEET9b-9e* contain a SPARK domain, and *FanSWEET9c* and *FanSWEET9d* also have the PKc_like superfamily. It can be seen that the structural domains of the *FanSWEET* protein family are conserved to a certain extent, but the number and distribution of the structural domains are somewhat differentiated.

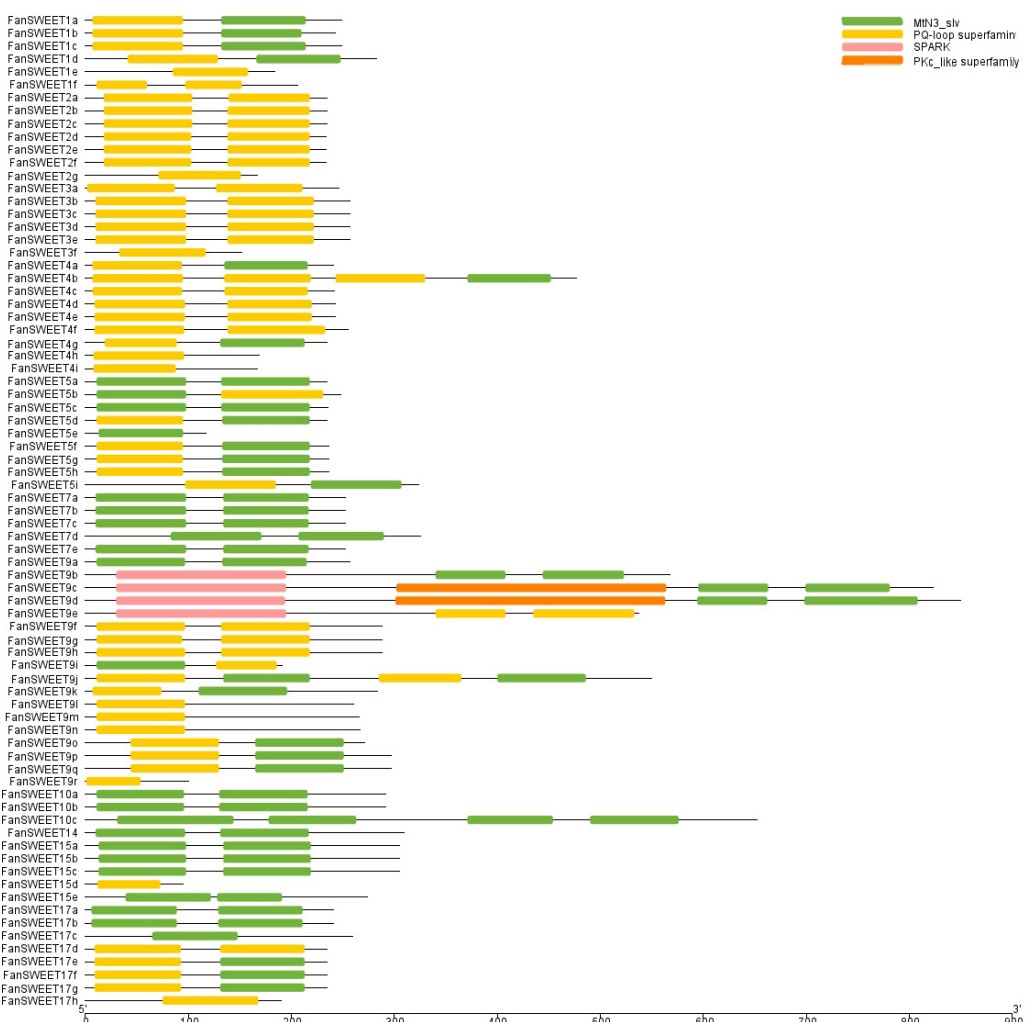

**Figure 3.** Conservative domains of SWEET family members of *F. × ananassa*. Different colored boxes represent different conserved domains.

The analysis of the motifs of *FanSWEET* members (Figure 4) showed that the members of this family contain 2 to 7 motifs, and Motif 1 and Motif 7 are mostly distributed at the N-terminal end, while Motif 2 and Motif 3 are mostly distributed at the C-terminal end. Motif 6 (fluorescent green) is the most conserved of all seven motifs because it is missing in only two *FanSWEETs* (*FanSWEET4i, 9r*). Fifty-three members of the *FanSWEETs* contain all seven conserved motifs, which is 68.83% of the total number. It can be seen (Figure 5) that G (glycine), P (proline), and Y (tyrosine) are highly conserved. I (isoleucine) and L (leucine) are present in all seven Motifs, and F (phenylalanine), G (glycine), and N (asparagine) are absent in Motif 2 only.

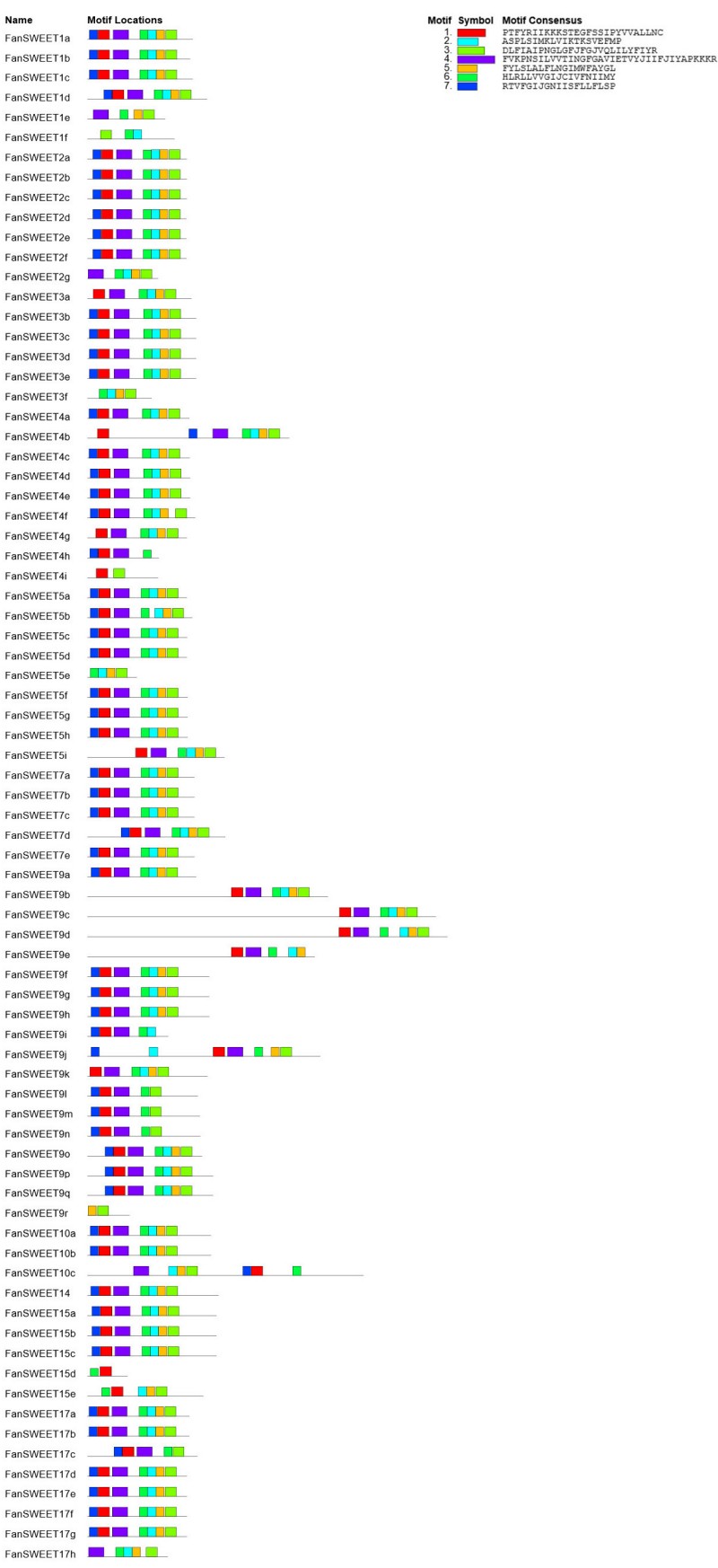

**Figure 4.** Conservative motif composition of SWEET family members of *F.* × *ananassa*. Different colored boxes represent different conserved protein motifs.

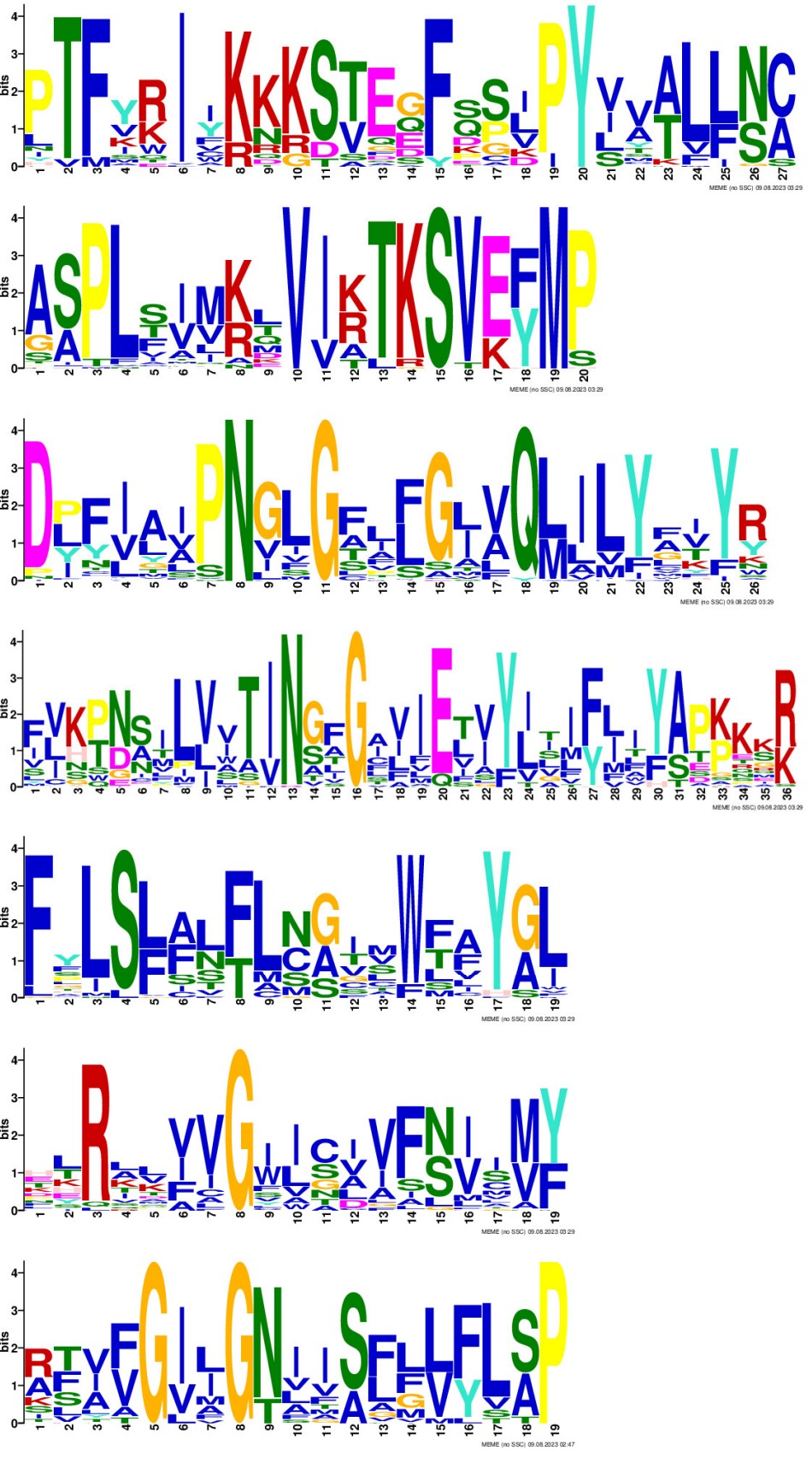

**Figure 5.** Identification of conservative motif sequences of SWEET family members of *F. × ananassa*. The bigger the letter of amino acid, the more conservative it is. The size of different amino acids in the same position is scaled according to their frequency.

### 3.5. Analysis of Gene Structure and Promoter Cis-Acting Elements of the SWEET Family of F. × ananassa

The number, location, and distribution of the exon–intron of the *FanSWEET* family are shown in Figure 6. There are thirty-five members without upstream and downstream regulatory regions. The number and length of introns and exons in the *FanSWEET* genes are extremely different. More than half of them (45, 58.44%) have six exons, and seventeen members have five exons (22.08%). *FanSWEET10c* contains the largest number of exons (13), while *FanSWEET4h*, *4i*, and *9r* contain the least number of exons (2).

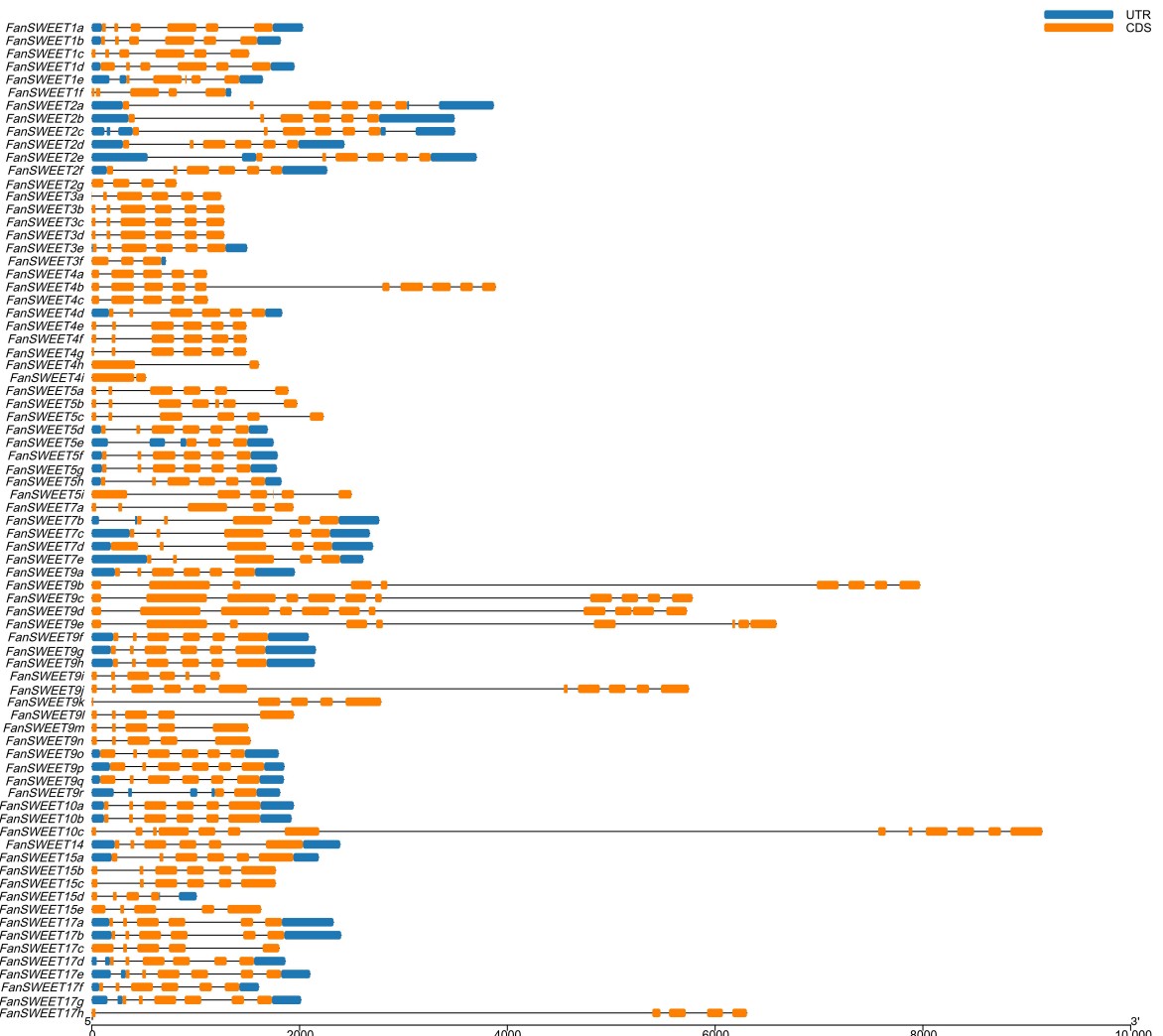

**Figure 6.** Genetic structure of *SWEET* family members of *F. × ananassa*. The orange box indicates the exon, the black line indicates the intron, and the blue box indicates the untranslated 5′- or 3′-region.

As shown in Figure 7, the four genes *FanSWEET 1a/15a/15b/15c* do not contain the 10 promoter elements looked up 2000 bp upstream. *FanSWEET3a* contains only one drought-responsive element; the majority of *FanSWEET* genes (0.78%) contain 10 to 27 cis-acting elements; and *FanSWEET2a* contains 39 cis-elements, the highest number of cis-acting elements in the family. Among the cis-acting elements identified in 'Camarosa' strawberry, light-responsive elements account for the largest proportion (46.26%) of the 10 elements. Circadian rhythm-responsive elements account for the smallest proportion, accounting for only 0.015% of the total number of elements. Among the phytohormone-responsive elements, MeJA-responsive elements were the most numerous (167), followed by abscisic-acid-responsive elements (119).

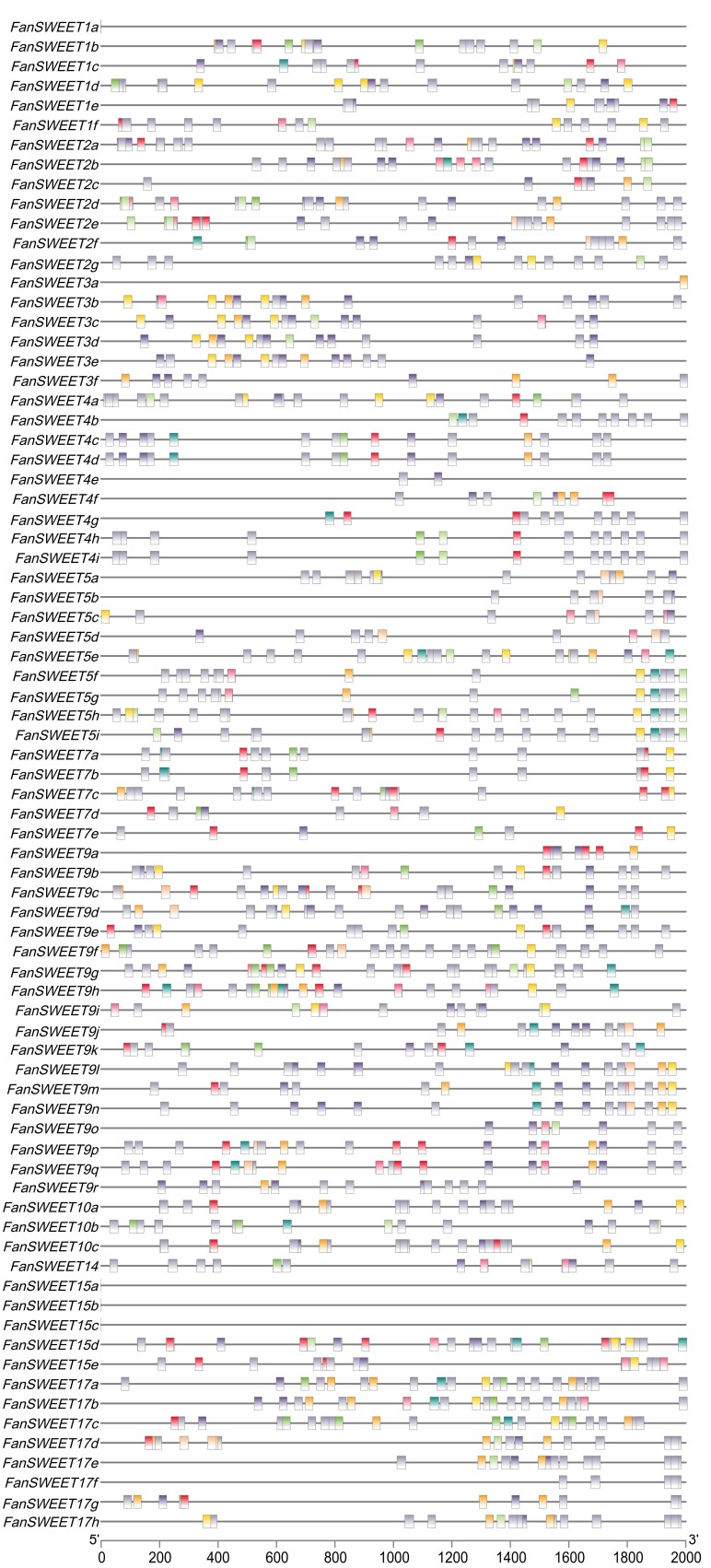

**Figure 7.** Homeopathic elements of *SWEET* family members of *F. × ananassa*. Different colored boxes represent different cis-acting elements.

*3.6. Analysis of FanSWEET Gene Expression*

3.6.1. Analysis of *FanSWEETs* Expression in Different Organs and Fruit Development Stages of Strawberry

By comparing the sequences of SWEET proteins in *A. thaliana* and *F. × ananassa*, 11 *FanSWEETs* (*FanSweet1a/2a/3b/4c/5a/7a/9b/10a/14/15a/17a*) with high similarity to the AtSWEET amino acid sequences were screened out for the determination of expression.

The expression of 11 *FanSWEET* genes in the fruits and leaves of the 'Yanli' strawberry was detected by RT-qPCR. The experimental results (Figure 8) showed that the expression patterns of *FanSWEET* genes in leaves and different development stages of fruits were quite different, and the expression level of *FanSWEET* genes in fruits was lower than that in leaves. Among them, the expression level of *FanSWEET10a/14/15a* was higher in the fruits (1.009, 1.006, and 1.012) and lower in the leaves (0.156, 0.006, and 0.064), while the other eight *FanSWEET* genes were highly expressed in the leaves and low in the fruits. This indicates that these genes may have different functions in the fruits and leaves. Among them, the expression of *FanSWEET17* in leaves is 19.05 times that in small green fruit and 150.59 times that in red fruits. The expression levels of all *FanSWEET* genes in strawberry fruits at the white stage are relatively low. With the gradual development of strawberry fruit, the expression of *FanSWEET4c/14/15a* decreased, while the expression of *FanSWEET1a/7a/9b/10a* first increased and then decreased. It is speculated that these *FanSWEET* genes participated in the process of sugar transportation and accumulation during fruit development. The highest expression level in the red fruit stage was *FanSWEET1a*, and the expression level in the smaller green fruit stage increased by 1.62 times. The lowest fruit expression level in the red fruit stage was *FanSWEET10a*, which decreased by 65.97 times in the small green fruit stage.

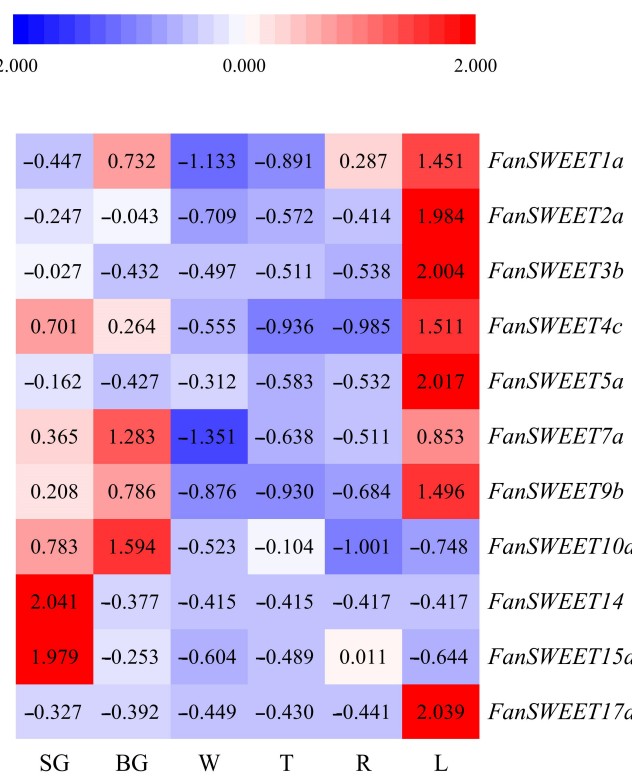

**Figure 8.** Expression patterns of the *FanSWEET* genes in leaves and different development stages of fruits of 'Yanli' strawberry. SG, BG, W, T, R, and L represent small green fruits, big green fruits, white fruits, turning fruits, red fruits, and leaves, respectively. The color grade indicates gene expression level: blue indicates low expression level, and red indicates high expression level.

### 3.6.2. Expression of *FanSWEETs* in Strawberry Fruits Treated with Exogenous Sugar

By spraying glucose, fructose, sucrose, and their mixed sugar on 'Yanli' strawberry plants, the expression of the *SWEET* genes in strawberry plants was examined. The results showed that spraying exogenous sugar changed the expression of the *FanSWEET* genes in fruits (Figure 9). Glucose treatment significantly reduced the expression of *FanSWEET1a/5a/7a/10a/15a/17a* and significantly increased *FanSWEET4c*. The expression level of *FanSWEET4c* significantly increased after exogenous fructose treatment at two concentrations, while the expression level of five *FanSWEET* genes (*FanSWEET5a/7a/10a/15a/17a*) decreased. However, the expression level of *FanSWEET1a* decreased by 35.986 times at low concentrations and increased by 1.284 times at high concentrations. Low-concentration treatment promoted the expression of *FanSWEET2a/14*, while high-concentration treatment inhibited the expression of these two genes. Under different concentrations of sucrose, the expression level of *FanSWEET1a* increased by 1.592 and 1.165 times, that of *FanSWEET5a/7a/15a/17a* significantly decreased, while that of *FanSWEET2a/9b/14* did not significantly change. Spraying exogenous mixed sugar reduced the expression level of six *FanSWEET* members (*FanSWEET1a/5a/7a/10a/15a/17a*), but increased the expression level of *FanSWEET4c*. The treatment of 0.2 mol·L$^{-1}$ mixed sugar highly induced *FanSWEET9b*, and the expression level was 4.407 times that of the control group. *FanSWEET2a* and *FanSWEET3b* were inhibited by low concentration and promoted by high concentration under glucose treatment. Under fructose treatment, *FanSWEET1a* was always expressed at low concentrations, and high concentration promoted expression, while *FanSWEET2b/3c/14* showed that low concentration promoted expression and high concentration inhibited expression. The expression of *FanSWEET3b/5a/10a/14* was inhibited under 0.1 mol·L$^{-1}$ sucrose treatment, but promoted under 0.2 mol·L$^{-1}$ sucrose treatment.

### 3.6.3. Expression of *FanSWEETs* in Strawberry Leaves Treated with Exogenous Sugar

Under the treatments of exogenous sugar, the expression of the *FanSWEET* genes in leaves generally decreased (Figure 10). The expression levels of six *FanSWEET* genes (*FanSWEET1a/2a/7a/9b/10a/14*) significantly decreased under different sugar treatments. This may be because the application of exogenous sugar leads to the accumulation of enough sugar in leaves, and the sugar transporters no longer played the role in transporting sugar. The expression level of *FanSWEET4c* was down-regulated by glucose, fructose, and sucrose alone and up-regulated by mixed sugar. Different from other sugar treatments, the expression of *FanSWEET3b/4c/5a/17a* increased under 0.1 MS treatment, which may play different roles in leaves. The expression level of *FanSWEET15a* was up-regulated by 1.205 times under 0.2 S treatment, and *FanSWEET17a* was up-regulated by 1.290 times under 0.1 MS treatment. Up-regulated genes (*FanSWEET3b/4c/5a/15a/17a*) were all caused by exogenous sucrose and mixed sugar spraying, which indicated that these genes were regulated by sucrose content. The expression of *FanSWEET3b/4c/15a* was inhibited at low concentration and promoted at high concentration. However, the expression levels of *FanSWEET3b/5a/17a* increased under low-concentration mixed sugar treatment and decreased under high-concentration treatment.

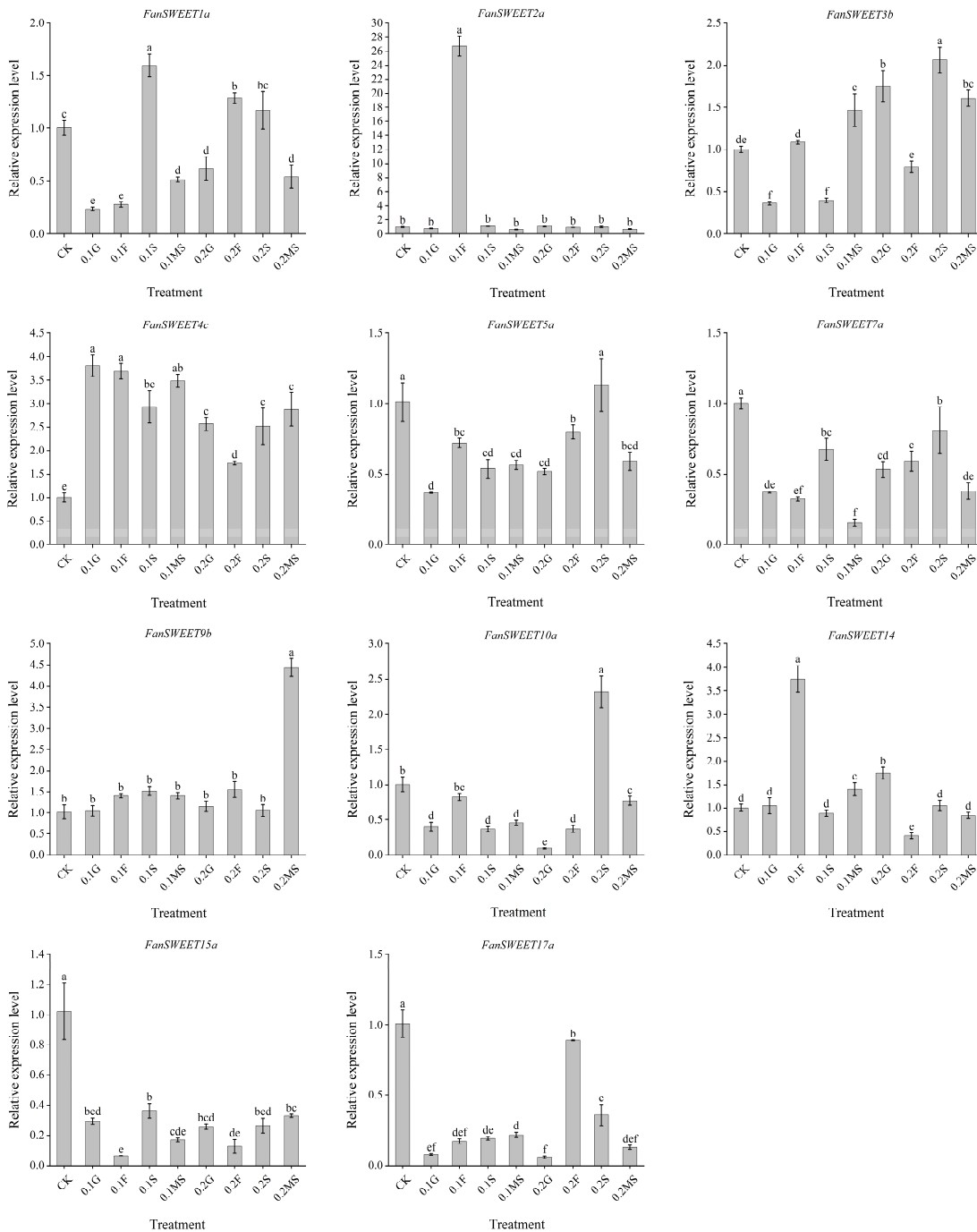

**Figure 9.** Expression of the *FanSWEET* genes in 'Yanli' strawberry fruits treated with exogenous sugar. CK is clear water control, with 0.1 G representing 0.1 mol·L$^{-1}$ glucose, 0.1 F representing 0.1 mol·L$^{-1}$ fructose, 0.1 S representing 0.1 mol·L$^{-1}$ sucrose, 0.1 MS representing 0.1 mol·L$^{-1}$ mixed sugar, 0.2 G representing 0.2 mol·L$^{-1}$ glucose, 0.2 F representing 0.2 mol·L$^{-1}$ fructose, 0.2 S representing 0.2 mol·L$^{-1}$ sucrose, and 0.2 MS representing 0.2 mol·L$^{-1}$ mixed sugar. The letters above the bars indicated the significant differences by student's *t*-test ($p < 0.05$). Three biological replicates were analyzed, and the error bars represented the SD.

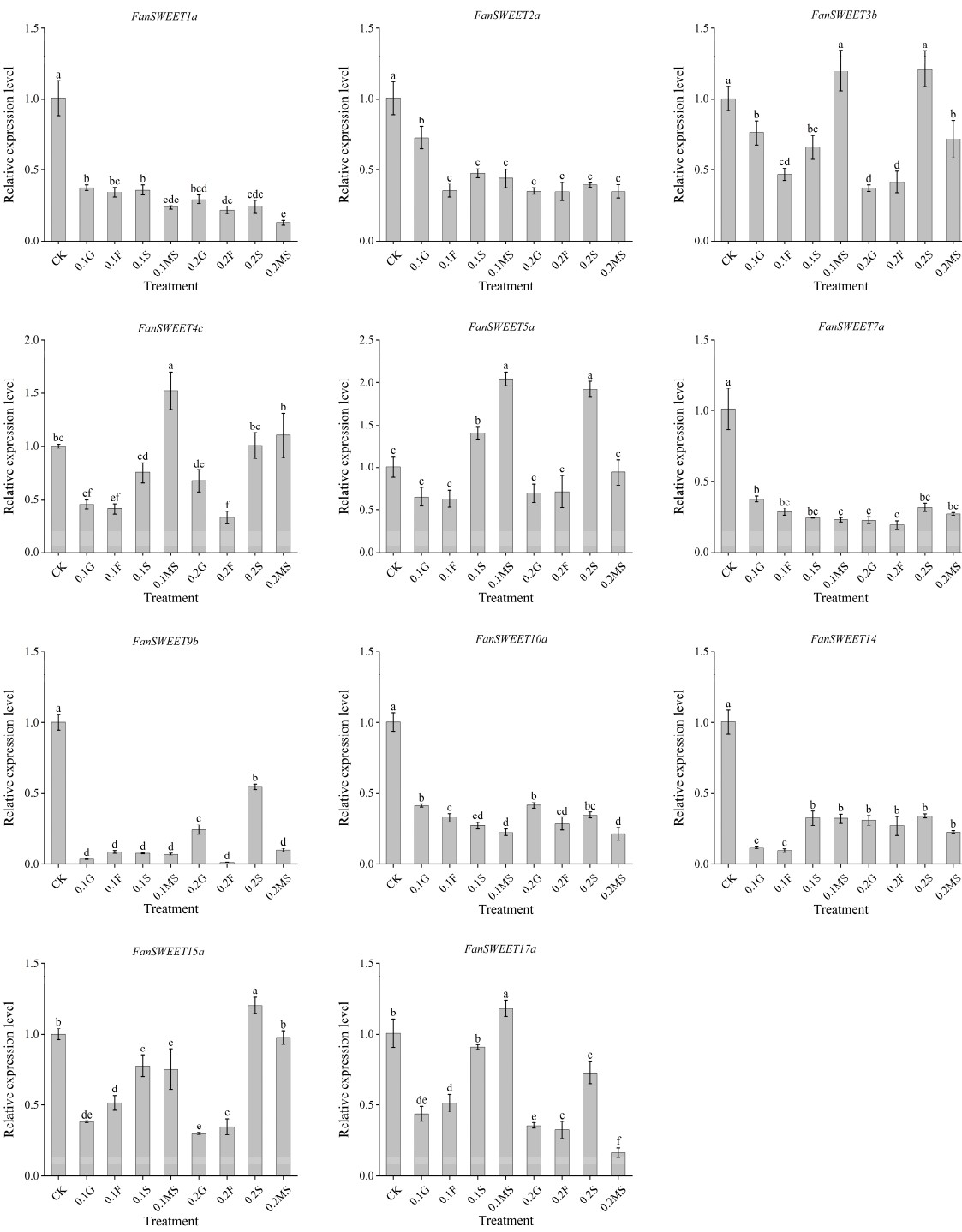

**Figure 10.** Expression of the *FanSWEET* genes in 'Yanli' strawberry leaves treated with exogenous sugar. CK is clear water control, with 0.1 G representing 0.1 mol·L$^{-1}$ glucose, 0.1 F representing 0.1 mol·L$^{-1}$ fructose, 0.1 S representing 0.1 mol·L$^{-1}$ sucrose, 0.1 MS representing 0.1 mol·L$^{-1}$ mixed sugar, 0.2 G representing 0.2 mol·L$^{-1}$ glucose, 0.2 F representing 0.2 mol·L$^{-1}$ fructose, 0.2 S representing 0.2 mol·L$^{-1}$ sucrose, and 0.2 MS representing 0.2 mol·L$^{-1}$ mixed sugar. The letters above the bars indicated the significant differences by student's *t*-test ($p < 0.05$). Three biological replicates were analyzed, and the error bars represented the SD.

## 4. Discussion

### 4.1. Bioinformatics Analysis of FanSWEET Gene Family

SWEET protein plays an important role in the growth and development of plants, which has been identified in many horticultural plants [4,47]. Research shows that 18 SWEET transporters were identified in pear [48]; 52 *SWEET* genes with high homology were blasted in soybean [49]. In this report, 77 members of the *FanSWEETs* protein family were found in the genome of octoploid *F. × ananassa*, and the number of *SWEET* genes was significantly higher than that of other species [10,16], which may be due to the difference of ploidy or usage methods. pI is an important parameter of protein, which is determined by the relative content of amino acid residues at different pH, and it affects the stability and physiological function of protein [50]. The pI of most *FanSWEET* members is greater than 7, while the pI of *FanSWEET1e/9l/9r/15a/15b/15c/15e/17h* is less than 7. However, the pI of *Arabidopsis* homologous genes *AtSWEET1/9/15* is greater than 7 [10]. This analysis leads us to speculate that their functions and modes of action are different from those of *Arabidopsis*. According to our analysis, most *FanSWEETs* have 7 TMHs, and a few members have 2 to 6 TMHs, implying that duplication and fusion of SWEETs may still be going on [51,52]. Knowing the subcellular localization information of protein can provide necessary help for us to infer the biological function of protein. This study predicts that *FanSWEETs* were distributed in the plasma membrane, vacuole, chloroplast, endoplasmic reticulum, and Golgi apparatus, indicating that they participate in various physiological functions in plants.

It is public knowledge that polyploidy is a crucial process in plant evolution, and many angiosperms have experienced polyploidy, which subsequently leads to gene replication in gene families [53,54]. As an octoploid species, *F. × ananassa* has a high probability of gene duplication. A total of 77 *FanSWEET* genes were distributed on all chromosomes except Chr1. Interestingly, we found that 76 gene fragments were duplicated, and tandem duplication clusters were observed on Chr 6-1 (*FanSWEET4f-FanSWEET4g*). These data suggest that the segmental duplication of genes in *F. × ananassa* resulted in gene family expansion, and we speculate that some genes may have functional redundancy or synergy. The membership in a clade can slightly define the substrate specificity of SWEETs [55]. According to the classification method of the phylogenetic tree of *A. thaliana* [10], 77 *FanSWEETs* were also divided into four subgroups, which is the same as other plants [56]. Based on the research on *Arabidopsis*, members in clade I (*FanSWEET1a-3f*) and II (*FanSWEET4a-7e*) may transport monosaccharides, those in clade III (*FanSWEET9a-15e*) are predominantly involved in sucrose uptake, and the proteins in clade IV (*FanSWEET17a-17h*) may mediate fructose transport [10,32,34,57].

The SWEET proteins belong to the MtN3_slv subfamily and the PQ-loop subfamily. The MtN3_slv subfamily is involved in glucose transport, while the PQ-loop subfamily is involved in amino acid transport [58]. Conservative domain analysis showed that most family members contained two MtN3_slv or PQ-loop superfamily domains. This indicated that the SWEET protein family of *F. × ananassa* was relatively conservative in evolution. Individual members (*FanSWEET9b-9e*) also contain SPARK and PKc_like domains, which play a role in signal transduction during plant–fungus symbiosis and catalytic transfer of amino acid [59]. The intron is involved in many important biological processes, such as mRNA output, transcription coupling, alternative splicing, gene expression regulation, and so on [60,61]. The analysis of gene structure shows that the number and length of introns and exons are quite different among *FanSWEET* genes, which may be the reason for functional diversity. Moreover, because the first two exons of the *FanSWEET* gene are very short, it is easy to be lost over time. Take *FanSWEET2g* for example, its first two exons are lost, leaving only four exons.

### 4.2. Expression of FanSWEET Genes in Strawberry

The evidence that the SWEET protein is involved in sugar accumulation in fruits has been reported in many articles. Ko et al. revealed that SlSWEET15 in tomatoes was involved in the unloading of sugar in fruit [62]. The expression of the *EjSWEET15* gene in

loquat may regulate the sugar accumulation process in mature fruit [63]. In this study, it was found that *FanSWEET1a/2a/3b/4c/5a/7a/9b/17a* were mainly expressed in leaves, while *FanSWEET10a/14/15a* were mainly expressed in fruits. Overall, the expression was low in fruits and high in leaves, which had obvious tissue specificity. That means they have various functions in different organs, as is evident from their expression patterns in other plant species [4,16,29,64,65].

Guo et al. found that *AnmSWEET5* and *AnmSWEET11* exhibited gradually down-regulated expression profiles during fruit development, especially in the early stages [66]. Recently, Yang et al. identified 19 *ZjSWEET* genes in jujube, and the expression levels of *ZjSWEETs* also fluctuated during fruit development [67]. From our research, we found that at the stage of small green fruit, *FanSWEET14* and *FanSWEET15a* were dominant expression genes. However, the expression of these genes gradually decreased with the development of fruits, showing a negative correlation with the accumulation of sugar in fruits. *FanSWEET2a/3b/5a/17a* were always expressed low during the fruit development of 'Yanli' strawberry, with little effect. The expression levels of *FanSWEET1a/7a/9b/10a* reached the highest in the big green fruit stage, and then rapidly decreased, indicating that they played an important role in the fruit expansion stage. In five stages of fruit development, the expression of *FanSWEETs* was higher in the small green fruit stage and the big green fruit stage, but lower in the white fruit stage, turning stage, and red fruit stage. The results indicated that *FanSWEETs* participate in regulating the process of strawberry fruit-ripening process, but their specific functions require further study.

*4.3. Effect of Exogenous Sugar Treatments on the Expression of FanSWEETs*

The expression of genes related to sugar metabolism in plants is affected by sugars. The expression of *CitSUT1* in mature leaf trays is inhibited by exogenous sucrose, glucose, mannose, and glucose analog 2-deoxyglucose [68]. Both *PlSUT2* and *PlSUT4* of peony can be induced by sucrose treatment [69]. The expression of *SWEETs* can also respond to their substrates [70]. However, the influence of exogenous sugar treatment on the *SWEET* gene family is lacking.

The sugar components in strawberry fruit are mainly composed of sucrose, glucose, and fructose, and the content of these three sugars can reach more than 90% of the total sugar content in mature fruit [71]. Therefore, in this study, these three sugars and their mixed sugars were selected to treat strawberry plants, and the effects of exogenous sugars on the expression of the *SWEET* genes in strawberries were analyzed.

Our findings demonstrate that exogenous sugar-spraying treatment changed the sugar content in strawberry and then changed the expression of *FanSWEETs*. The expression levels of *FanSWEET2a* (26.717) and *FanSWEET14* (3.755) in fruits were significantly increased by spraying $0.1 \text{ mol·L}^{-1}$ fructose. But the expression levels of one of these two genes were not significantly changed (0.944), and the other was significantly decreased (0.408) by spraying $0.2 \text{ mol·L}^{-1}$ fructose. The results showed that these two genes were sensitive to the fructose content in fruits. On the contrary, the expression level of *FanSWEET9b* was significantly up-regulated under $0.2 \text{ mol·L}^{-1}$ mixed sugar (4.469). This is obviously different from the reaction of other genes to exogenous sugar, and the cause of this phenomenon need to be further studied.

Sugar acting on leaves can also promote or inhibit the expression of *FanSWEETs* in leaves. The expression levels of *FanSWEET1a/2a/7a/9b/10a/14* were all decreased under exogenous sugar treatment. Similar findings were found in the study of tomatoes [18]. *FanSWEET5a* was only sensitive to sucrose and mixed sugar, and *FanSWEET17a* showed the phenomenon that low concentration promotes expression and high concentration inhibits expression under different concentrations of mixed sugar.

In this article, a relatively comprehensive bioinformatics analysis of the *FanSWEET* gene family was carried out, which provides a basis for further understanding the potential functions and characteristics of *FanSWEET* genes. As sugar transporters, these genes play essential roles in the growth and development of strawberry as well as the response to

exogenous sugar. The understanding of their exact functions, however, is still incomplete. Thus, in order to improve fruit quality, in-depth functional verification research is needed to provide valuable insights.

## 5. Conclusions

In this study, 77 members of *FanSWEET* genes were identified by systematic bioinformatics analysis, and their physical and chemical properties, phylogenetic analysis, conserved domains, conserved motifs, gene structure, and chromosome location were analyzed. A total of 11 *FanSWEET* genes were selected for further expression analysis by comparing them with *AtSWEET* gene sequences of *A. thaliana*. Among them, 3 *FanSWEETs* were highly expressed in fruits, and 8 *FanSWEETs* were highly expressed in leaves. Further experiments showed that the transcriptions of these genes were influenced by exogenous sugar. Glucose treatments decreased the expression of *FanSWEET1a/5a/7a/10a/15a/17a* and increased the expression of *FanSWEET4c/9b/14*. Under fructose treatments, the expression of *FanSWEET5a/7a/10a/15a/17a* decreased, while the expression of *FanSWEET4c/9b* increased. Sucrose treatments decreased the expression of *FanSWEET7a/15a/17a* but increased the expression of *FanSWEET1a/2a/4c/9b*. Mixed sugar treatments decreased the expression of *FanSWEET1a/2a/5a/7a/10a/15a/17a* and increased the expression of *FanSWEET3b/4c/9a/14*. *FanSWEET* genes in leaves were mainly increased by sucrose and mixed sugar treatment but were decreased by glucose and fructose treatment. This study provided a basis for further research on the *FanSWEET* gene family of strawberry and *SWEET* genes in other species.

**Supplementary Materials:** The following supporting information can be downloaded at: https://www.mdpi.com/article/10.3390/horticulturae10020191/s1, Table S1: Primer sequences in RT-qPCR.

**Author Contributions:** Conceptualization, R.T. and H.L.; methodology, R.T.; software, R.T. and J.L.; validation, J.X. and Z.X.; writing—original draft preparation, R.T.; resources, H.L.; writing—review and editing, R.T. and H.L.; project administration, H.L. All authors have read and agreed to the published version of the manuscript.

**Funding:** This work was financially supported by the Liaoning Key Agricultural Project (2023JH1/10200003), the Liaoning Key R&D Program (2020JH2/10200032), the National Key R&D Program of China (2019YFD1000200), and the Shenyang Science and Technology Mission Program (23-410-2-04).

**Data Availability Statement:** Data are contained within the article and supplementary materials.

**Conflicts of Interest:** The authors declare no conflict of interest.

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
