# Peer review of "Genome-Wide Identification and Expression Analysis of SWEET Gene Family in Strawberry"

_horticulturae, doi:10.3390/horticulturae10020191_

Round 1

Reviewer 1 Report

Comments and Suggestions for Authors

Need data/table on effect of exogenous sugar on FanSWEET genes  expression.

Reviewer 2 Report

Comments and Suggestions for Authors

The authors conducted an extensive investigation of the SWEET gene family in strawberries in this study. However, it is crucial to note that a substantial portion of the results presented in this manuscript relies on in silico analysis. Additional information regarding the wet lab experiments, specifically those conducted using RT-qPCR, is necessary to enhance the comprehensiveness and reliability of the findings.

1. Provide expansion for many software used such as MEGA.

2. check all the links are working. For example, there is an extra space in the weblink for MEME in line number 137. So, this link is not working.

3. why 11 FanSWEET genes were selected based on homologous to Arabidopsis genes. Is there any particular reason?

4. check the sentences. “Using MEGA11 software, the identified strawberry  SWEET members were ClustalW compared with the AtSWEET amino acid sequences, respectively.” It is not understandable. Correct it. Check the complete manuscript for such errors.

5. “with a spray can” in line 161. It should be cane.

6. In Section 2.6, at what specific time point after the sugar treatment were the samples collected for expression studies? The authors noted that "The exogenous sugar-treated samples were collected at the red fruiting stage.” However, it is important to acknowledge that this timing may vary for each plant, resulting in variations in the number of days of treatment given to each sample.

7. what reference gene/ housekeeping gene was used for the expression analysis? how it was selected?

8. The below sentence is been duplicated in the manuscript. Dele one. “The reaction system was as follows: 0.5 μL of template cDNA, 0.5 μL of each 175 upstream and downstream primer (Table 2), 5 μL of SYBR dye, and 10 μL of ddH2O”

9. it is not 5 μL of SYBR dye. It should be 5 μL of SYBR Master mix as it has nucleotides, polymerase and dye.

10. Total RT-qPCR volume comes 16.5 µl based on the manuscript? Is it you used a total volume of 16.5 µl

11. How many biological replications were used for expression studies?

12. provide expansion for the last column in Table 3.

13. why three different sugars were used for expression analysis? provide details in introduction or discussion more.

Comments on the Quality of English Language

Extensive editing is required

Reviewer 3 Report

Comments and Suggestions for Authors

Dear Authors,

allow me to make a few suggestions for corrections and additions to the text.

- SWEET;  The Sugars Will Eventually be Exported Transporters

- line 130;  10-10

- line 132; 100 kb

- line 149; please shortly describe the genotype Yanli

- line 151 ...ShenYang  (Li....

- Table 1; it is necessary to include the abbreviations explanation

- lines 169-180; for chemicals and devices, it is necessary to indicate the manufacturers

- lines 169-180; the Rt-qPCR protocol must content the concentrations of individual components (cDNA, primers etc.) not volumes

- line 169; please indicate the reference of CTAB method

- lines 172 and 174 are duplicated,

- line 174; ddH2O

- please include the reference gene used for internal control

- please include the thermal and time profile of qPCR

-line 178; please include the reference for the relative expression calculation

- line 180; please indicate the reference for the statistical analyses

- Tables 2 and 3; I suggest to include these tables to supplements 

Reviewer 4 Report

Comments and Suggestions for Authors

The sugar transporter proteins play a crucial role in plant growth and development. This study systematically investigated the SWEETs family in octoploid strawberry (Fragaria × ananassa Duch.) and showed their expression profiles in various tissues and fruit development stages. The topic is interesting and may attract attentions in strawberry breeding and research communities. However, the authors focus more on the bioinformatical analysis but didn’t perform sound experiments to address biological or agricultural questions. I didn’t see the authors clearly stated the biological question(s) nor the followed hypothesis and prediction. I didn’t see any conclusions derived from the results, which are basically the description of the authors observations of the SWEETs family based on the bioinformatical analysis. Although there are gene expression analysis of the SWEETs gene family and an exogenous sugar treatment assay to test its impacts on, still, I didn’t see any take home messages clearly stated in the manuscript. Taken together, I’m not convinced the current manuscript deserves to be published as the overall merits are low.
